# Collagen Network Formation in In Vitro Models of Musculocontractural Ehlers–Danlos Syndrome

**DOI:** 10.3390/genes14020308

**Published:** 2023-01-24

**Authors:** Ayana Hashimoto, Takuya Hirose, Kohei Hashimoto, Shuji Mizumoto, Yuko Nitahara-Kasahara, Shota Saka, Takahiro Yoshizawa, Takashi Okada, Shuhei Yamada, Tomoki Kosho, Takafumi Watanabe, Shinji Miyata, Yoshihiro Nomura

**Affiliations:** 1Graduate School of Agriculture, Tokyo University of Agriculture and Technology, 3-5-8 Saiwaicho, Fuchu, Tokyo 183-8509, Japan; 2Laboratory of Veterinary Anatomy, School of Veterinary Medicine, Rakuno Gakuen University, Ebetsu 069-8501, Hokkaido, Japan; 3Department of Pathobiochemistry, Faculty of Pharmacy, Meijo University, Nagoya 468-8503, Aichi, Japan; 4Division of Molecular and Medical Genetics, Center for Gene and Cell Therapy, The Institute of Medical Science, The University of Tokyo, Minato-ku, Tokyo 108-8639, Japan; 5Division of Animal Research, Research Center for Advanced Science and Technology, Shinshu University, Matsumoto 390-8621, Nagano, Japan; 6Department of Medical Genetics, Shinshu University School of Medicine, Matsumoto 390-8621, Nagano, Japan; 7Center for Medical Genetics, Shinshu University Hospital, Matsumoto 390-8621, Nagano, Japan; 8Division of Clinical Sequencing, Shinshu University School of Medicine, Matsumoto 390-8621, Nagano, Japan; 9Research Center for Supports to Advanced Science, Matsumoto 390-8621, Nagano, Japan

**Keywords:** Ehlers–Danlos syndrome, decorin, collagen, dermatan sulfate proteoglycan, fibrillogenesis, carbohydrate sulfotransferase 14, mcEDS-*CHST14*

## Abstract

Loss-of-function mutations in *carbohydrate sulfotransferase 14* (*CHST14*) cause musculocontractural Ehlers–Danlos syndrome-*CHST14* (mcEDS-*CHST14*), characterized by multiple congenital malformations and progressive connective tissue fragility-related manifestations in the cutaneous, skeletal, cardiovascular, visceral and ocular system. The replacement of dermatan sulfate chains on decorin proteoglycan with chondroitin sulfate chains is proposed to lead to the disorganization of collagen networks in the skin. However, the pathogenic mechanisms of mcEDS-*CHST14* are not fully understood, partly due to the lack of in vitro models of this disease. In the present study, we established in vitro models of fibroblast-mediated collagen network formation that recapacitate mcEDS-*CHST14* pathology. Electron microscopy analysis of mcEDS-*CHST14*-mimicking collagen gels revealed an impaired fibrillar organization that resulted in weaker mechanical strength of the gels. The addition of decorin isolated from patients with mcEDS-*CHST14* and *Chst14*^−/−^ mice disturbed the assembly of collagen fibrils in vitro compared to control decorin. Our study may provide useful in vitro models of mcEDS-*CHST14* to elucidate the pathomechanism of this disease.

## 1. Introduction

Ehlers–Danlos syndrome (EDS) comprises a clinically and genetically heterogeneous group of heritable connective tissue disorders characterized by joint hypermobility, skin hyperextensibility, and tissue fragility [1,2]. Currently, EDS is classified into 14 subtypes based on clinical, molecular and biochemical features according to the 2017 International Classification and more recent updates [3,4]. Dominant negative effects or haploinsufficiency of mutant procollagen α-chain genes or deficiency of collagen-processing enzymes have been identified as the basis for various types of EDS [4,5].

Musculocontractural EDS-CHST14 (mcEDS-CHST14) is a recently delineated subtype of EDS caused by biallelic loss-of-function mutations in CHST14 [6,7,8,9]. Clinical characteristics of mcEDS-CHST14 include multiple malformations (e.g., craniofacial features, multiple congenital contractures, ocular and visceral malformations) and progressive fragility-related multisystem manifestations (e.g., skin hyperextensibility and fragility, joint hypermobility, progressive spinal and foot deformities, large subcutaneous hematomas, and visceral ruptures) [6,10]. CHST14 encodes dermatan 4-O-sulfotransferase 1, a Golgi-resident enzyme essential for the biosynthesis of dermatan sulfate (DS) chains [11,12]. DS, a type of linear sulfated polysaccharide called glycosaminoglycans (GAGs), is built up by repeating disaccharide units of iduronic acid (IdoA) and N-acetylgalactosamine (GalNAc). Dermatan is formed by converting a precursor GAG called chondroitin comprising glucuronic acid (GlcA) and GalNAc [13]. Epimerization of GlcA in chondroitin to IdoA by dermatan sulfate epimerase (DSE) and subsequent sulfation of the C4-position of GalNAc by CHST14 completes the biosynthesis of DS chains. DS chains are covalently attached to specific core proteins to form DS-proteoglycans widely distributed in the extracellular matrix. The epimerization reaction in vitro is reversible and favors GlcA formation, whereas the sulfation by CHST14 is irreversible [14]. CHST14 and DSE form complexes in the Golgi apparatus that generate long stretches of IdoA contain disaccharides [15]. Therefore, patients with mcEDS-CHST14 lacking functional CHST14 exhibit marked loss of DS with an excess amount of chondroitin sulfate (CS), which results from the replacement of DS with CS in their connective tissues, including skin [7]. 

Patients with mcEDS-*CHST14* and mice model of mcEDS-*CHST14* deficient in *Chst14* (*Chst14*^−/−^) shows misoriented collagen fibrils, disorganized collagen fibers, and decreased skin tensile strength [7,8,16,17]. It has been proposed that the disturbed assembly of collagen networks is caused by the structural alteration of GAG chains on decorin [8,16,18]. CS chains are linear and extend from the outer surface of collagen fibrils, whereas DS chains are curved and maintain close contact with attached collagen fibrils [16]. Decorin is a major DS-proteoglycan in the connective tissues and plays an essential role in collagen fibrillogenesis [19]. However, the molecular consequences of *CHST14* mutations in the pathology of mcEDS-*CHST14* are not fully understood, partly due to the lack of in vitro models that recapitulate pathogenic alterations in collagen network formation. 

Fibroblasts cultured within free-floating type I collagen gels reorganize the surrounding collagen fibrils into a more dense and compact arrangement [20,21,22]. This fibroblast-mediated remodeling of collagen network provides an effective in vitro model for studying skin morphogenesis and pathogenesis. In the present study, we aimed to evaluate the impacts of fibroblasts derived from patients with mcEDS-*CHST14* on remodeling type I collagen network in vitro. We also examined the effects of decorin isolated from patients’ fibroblasts on fibrillar organization. We revealed an impaired collagen assembly in the human and mouse models of mcEDS-*CHST14* that emphasize the usefulness of the in vitro pathological model to elucidate the pathomechanism of this disease.

## 2. Materials and Methods

### 2.1. Animals

*Chst14*^−/−^ mice were generated as described previously [23,24]. All experimental procedures were approved by the Experimental Animal Care and Use Committee at Tokyo University of Agriculture and Technology, the National Center of Neurology and Psychiatry), Nippon Medical School, and Shinshu University. Mice were housed in a micro isolator, at 23 ± 2 °C, with constant humidity and a 12 h light/dark cycle. Mice had free access to tap water and standard mouse chow.

### 2.2. Primary Fibroblast Culture

The dermal fibroblasts were established from skin specimens obtained through a standard procedure. We used skin fibroblasts from two patients with mcEDS-*CHST14* (NM_130468.3) (patient 3, a 32-year-old man: [c.842C>T];[c.842C>T],p.[Pro281Leu];[Pro281Leu]; patient 6, a 4-year-old girl: [c.842C>T];[c.878A>G],p.[Pro281Leu];[p.Tyr293Cys]) as described previously [7]. Normal human fibroblasts from healthy individuals were purchased from Japan Health Sciences Foundation. Human and mouse dermal fibroblasts were cultured in Dulbecco’s modified Eagle’s medium (DMEM) with 10% heat-inactivated fetal bovine serum (FBS), 100 U/mL penicillin, 100 U/mL streptomycin. 

### 2.3. Quantification of DS Chains

GAGs in the cultured medium of fibroblasts were analyzed as described previously [7]. Briefly, samples were individually digested with chondroitinase AC (Seikagaku Co. Tokyo, Japan) or B (IBEX Pharmaceuticals, Montreal, QC, Canada) for the specific analysis of CS or DS moiety, respectively. After conjugation with 2-aminobenzamide, the labeled disaccharides were separated by anion-exchange HPLC on a PA-G silica column (4.6 × 150 mm; YMC Co. Kyoto, Japan) and monitored using a fluorometric detector. The identification and quantification of the resulting disaccharides were achieved by comparisons with the elution positions of DS-derived authentic unsaturated disaccharides. The amounts of DS disaccharides were normalized by protein levels.

### 2.4. Preparation of Fibroblast-Embedded Collagen Gel

The collagen gels were prepared as described previously with minor modifications [22]. Briefly, 1 × 10^6^ dermal fibroblasts were suspended in 4 mL of DMEM containing FBS, sodium bicarbonate, penicillin, and streptomycin. Then, 4 mL of the cell suspension was mixed with 2 mL of bovine type I collagen acidic solution (3 mg/mL, pH 3.0, Nitta Gelatin Co., Osaka, Japan, #IAC-30). Then, 5 mL of the mixture was placed in a 6-well plastic dish and incubated, at 37 °C, for allowing gelation of collagen. The resultant collagen gels containing fibroblasts were detached from the surrounding brim of plastic dishes and maintained, at 37 °C, in air supplemented with 5% CO_2_. The gel contraction was evaluated by measuring the diameter of the gels at the indicated time. We used the 0.1% collagen gels for gel contraction assay and transmission electron microscopy analysis and 0.22% collagen gels for measuring the breaking strength and scanning electron microscopy analysis.

### 2.5. Transmission Electron Microscopy

Transmission electron microscopy was performed as described previously [16,17]. Briefly, the collagen gels were prefixed in 3% glutaraldehyde in 0.1 M phosphate buffer (pH 7.4) for 2 h, at room temperature, and postfixed in 1% osmium tetroxide in phosphate buffer for 1 h, at room temperature. After washing with water, gels were dehydrated in ethanol and transferred to QY-1 (Nisshin EM, Tokyo, Japan). Samples were embedded in the epoxy-resin mixture (Nisshin EM), cut into ultrathin sections, and observed by transmission electron microscopy (JEM-1220; JEOL, Tokyo, Japan).

### 2.6. Scanning Electron Microscopy

Scanning electron microscopy was performed as described previously [17]. Briefly, the collagen gels were prefixed in 3% glutaraldehyde in phosphate buffer for 2 h, at room temperature, and postfixed in 1% osmium tetroxide in phosphate buffer for 1 h, at room temperature. After washing with phosphate buffer, the gels were incubated with 1.0% tannic acid in water for 30 min. The gels were again fixed in 1% osmium tetroxide for 1 h, at room temperature, and washed with phosphate buffer. The samples were then dried by the t-butyl alcohol freeze-drying method, mounted on metal stubs, coated with platina using an ion sputter (JUC-5000; JEOL), and observed under a scanning electron microscope (JSM-5200; JEOL).

### 2.7. Mechanical Strength Measurement of Collagen Gel

Mechanical strength of the collagen gel was analyzed using a creep meter (RE-33005, Yamaden) as described previously [25]. The breaking strength was measured in 60% humidity, at 25 °C, using a cylindrical probe (diameter 5 mm) moving into the gel at a speed of 5 mm/min. The penetration of the probe was stopped halfway in the whole thickness of the gel specimen. The peak top of the stress–strain curve was defined as the breaking point of a gel. The measurement was repeated 8 times for each gel.

### 2.8. Isolation of Decorin

Human decorin was isolated from the cultured medium of normal and the patient’s fibroblasts. Serum-free cultured medium (700–900 mL) was collected, freeze-dried, and dissolved in the extraction buffer (7 M urea, 50 mM Tris-HCl (pH 7.4), 0.1 M NaCl, and protease inhibitors). The extract was applied to a DEAE-Toyopearl 650 M column (Tosoh Corp., Tokyo, Japan) equilibrated with the extraction buffer. Decorin was eluted with stepwise increases of 0.3, 0.5, and 1.0 M NaCl. Fractionated samples were precipitated with ethanol and dissolved in 0.4 mL of water. For mouse decorin, skin powder was delipidated by mixing with ethanol and extracted with 4 M guanidine hydrochloride solution containing 50 mM Tris-HCl (pH 7.4), 0.1 M NaCl, 5 mM benzamidine hydrochloride, and 10 mM EDTA, at 4 °C, for 72 h with rotating. After centrifugation, the supernatant was dialyzed against the extraction buffer and applied to a DEAE column as described. 

To detect decorin, we separated samples by acrylamide gel electrophoresis, transferred them onto a polyvinylidene difluoride membrane (Millipore, Burlington, MA, USA), and incubated them with the following primary antibodies: anti-human decorin (mouse IgG1, 1:1000, R&D #115402), and anti-mouse decorin (goat IgG, 1:1000, R&D #AF1060). After washing, the membranes were incubated with the appropriate peroxidase-labeled secondary antibodies and developed using a chemiluminescent peroxidase substrate (Millipore). For human decorin, the 0.5 M NaCl fraction was collected, and for mouse decorin, the 0.3 M and 0.5 M NaCl fractions were pooled and used for subsequent experiments.

### 2.9. Chondroitinase Digestion of Decorin

Skin lysate obtained from *Chst14*^+/+^ and *Chst14*^−/−^ mice were digested with chondroitinase ABC (Seikagaku Co.) or chondroitinase B (IBEX Pharmaceuticals). The digested and undigested lysates were subjected to immunoblotting to detect decorin.

### 2.10. Decorin-Mediated Fibrillar Organization of Type I Collagen Gels

Isolated decorin was quantified by a direct enzyme-linked immunosorbent assay using the anti-decorin antibodies. Decorin was incubated with bovine type I collagen acidic solution (collagen:decorin = 500:1 (*w/w*)) for 2 h in a neutral solution. The resultant collagen gels were analyzed by scanning electron microscopy as described.

### 2.11. Statistical Analysis

All data are presented as the mean ± SE. An unpaired, two-tailed Student’s *t*-test was used to compare results from two groups. One-way ANOVA with Tukey–Kramer’s test was used for multiple-comparison test. A *p*-value < 0.05 was considered to be significant. Details of statistical analyses, including the statistical tests used and *p* values, may be found in the relevant figures and figure legends.

## 3. Results

### 3.1. Effect of Fibroblasts from mcEDS-CHST14 Patients on Contraction, Fibrillar Organization, and Mechanics of Type I Collagen Gels

We used dermal fibroblast derived from two patients with mcEDS-*CHST14* whose pathology was reported in a previous study [7]. Similarly to patient 3, whose GAGs were previously analyzed [7], the cultured medium of fibroblast from patient 6 contained CS but not DS (Appendix A). We then examined the effects of fibroblasts from patients with mcEDS-*CHST14* on the collagen gel contraction in vitro. Culturing control fibroblasts in a free-floating collagen gel resulted in a contraction of the gel in a time-dependent manner (Figure 1a,b). By measuring the diameter of the gel at different time points, we found that contraction of the collagen gel mainly occurred within the first 6 h under our experimental conditions. No gel shrinkage was observed in the absence of fibroblasts. The collagen gels cultured with two patient-derived fibroblasts showed significantly larger gel diameters and delayed contraction than that with the healthy control at 1 and 3 h (Figure 1a,b). After 24 h, the diameters of the gels were comparable regardless of the origin of the fibroblasts, suggesting that patient-derived fibroblasts delayed the progression of the initial phase of collagen gel contraction.

We observed fibrillar organization in the collagen gel by transmission electron microscopy analysis (Figure 2a). The assembly of collagen fibrils was evaluated by counting fibrils attached to other fibrils in the transverse sections. Three days after culturing with control fibroblasts, multiple fibrils were assembled to form collagen fibers (Figure 2a). In contrast, the assembly of fibrils was less frequent when cultured with patients-derived fibroblasts. The percentage of assembled fibrils in the total fibrils was 68.2% (n = 36 fibrils) in the control gel, and it was 38.9% and 50.0% in the two patient-mimicking gels, respectively (n = 36 and 16 fibrils for patient 3 and 6, respectively). The fibrillar organization of the gels was further examined using scanning electron microscopy (Figure 2b). The diameter size of fibrils was significantly increased by co-culturing with fibroblasts (Figure 2c). Although there is no apparent difference in the mean diameter of fibrils between control and patient-mimicking, the diameter size distribution of fibril was different between two groups. The diameter of fibrils in the control gels appeared more homogeneous than those in patient-mimicking gels (Figure 2d). These results suggest that patient-derived fibroblasts disturbed fibrillar organization of the collagen gels in vitro.

We investigated mechanical properties of the collagen gels by measuring the breaking strength using a creep meter. The stress–strain curves of the collagen gels indicated that the breaking strength of the collagen gel was markedly increased in the presence of control and patient-derived fibroblasts compared with fibroblast-free gels (Figure 3a). We found that the patient-mimicking gels reached the failure points at a lower stress than the control gels. Consistently, the patient-mimicking gels showed significantly lower maximum stress than the control gels (Figure 3b). These data indicated that the impaired fibrillar organization in patient-mimicking gels resulted in weaker mechanical strength of the gels than the control gels.

### 3.2. Effect of Decorin Isolated from Normal or Patient’s Fibroblasts on Type I Collagen Gel Properties

Various factors are involved in the fibroblast-mediated collagen gel contraction, making it challenging to identify the molecules that regulate the collagen fibrillar organization. We sought to examine whether the disturbed collagen network in the gels co-cultured with patient-derived fibroblasts was due to abnormalities in decorin. To this end, we isolated decorin from the cultured medium of control and the patient’s fibroblasts and investigated its effects on collagen fibrillogenesis in vitro. Anion exchange chromatography followed by immunoblot analysis indicated that decorin was highly enriched in the 0.5 M NaCl elution fraction (Figure 4a). The apparent molecular weight of decorin was undistinguishable between normal and patient fibroblasts. 

Type I collagen was incubated with or without decorin (collagen:decorin = 500:1 (*w/w*)) for 2 h in a neutral solution, and the resultant collagen gels were analyzed by scanning electron microscopy (Figure 4b and Appendix A). In the absence of decorin, the majority of collagen fiber was composed of two or three fibrils. Thicker collagen fibers were formed in the presence of decorin derived from normal fibroblasts, confirming the instructive roles of decorin in collagen fibrillar organization. Observation at higher magnification visualized GAG chains of decorin attached to collagen fibrils (Figure 4c and Appendix A). GAG chains of control decorin were evenly spaced and vertically oriented on the collagen fibers. In contrast, GAG chains of patient-derived decorin were misoriented and did not tightly attach to collagen fibrils. The diameter of collagen fibers was significantly smaller in the collagen gels incubated with patient-derived decorin than normal decorin (Figure 4d). We also found that the diameter of each fibril was smaller when incubated with patient-derived decorin compared to the control (Figure 4e). The diameter of fibrils was more variable in the collagen gel incubated with patient-derived decorin than with controls (Figure 4f). These results directly demonstrated an impaired function of patient-derived decorin in fibrillar organization of type I collagen.

### 3.3. Effect of Fibroblasts and Decorin Derived from Chst14-Deficient Mice on the Fibrillar Organization of Type I Collagen Gels

A mcEDS-*CHST14* model mouse lacking functional CHST14 protein has been established recently. We examined the effect of fibroblasts from *Chst14*-deficient mice on the fibrillar organization of type I collagen gels in vitro. Scanning electron microscopy analysis showed the assembly of collagen fibers composed of multiple fibrils (Figure 5a,b and Appendix A). The diameter of fibrils was markedly increased by co-culturing with fibroblasts compared with the cell-free condition (Figure 5a,b). We found that the diameter of fibrils was smaller in the collagen gels cultured with fibroblasts derived from *Chst14*^−/−^ mice compared to those from *Chst14*^+/+^ and *Chst14*^+/−^ mice (Figure 5c). Higher magnification views indicated that GAG chains were not tightly aligned on fibrils in *Chst14*-deficient condition (Figure 5b).

We digested the skin lysate with two chondroitinases to investigate GAG modification on decorin. Chondroitinase ABC degrades both DS and CS chains, whereas chondroitinase B acts only on DS. Decorin from *Chst14*^+/+^ mice was shifted to lower molecular weight after digestion with chondroitinase ABC and chondroitinase B (Figure 6a). In contrast, decorin from *Chst14*^−/−^ mice was partially resistant to chondroitinase B but was digested by chondroitinase ABC. This result indicated that DS chains on decorin were markedly reduced and replaced by CS chains in *Chst14*^−/−^ mice, as previously reported in patients with mcEDS-*CHST14* [7]. We isolated decorin from cultured media of mouse fibroblast using anion exchange chromatography (Figure 6b) and examined its effect on the fibrillar organization of type I collagen gels. Again, we found that the collagen gels incubated with decorin isolated from two *Chst14*^−/−^ mice had significantly smaller fibril diameters compared to decorin from *Chst14*^+/+^ and *Chst14*^+/−^ mice (Figure 6c,d). These results indicate that fibroblast- and decorin-mediated fibrillar organization of collagen are impaired by functional loss of CHST14 in both humans and mice.

## 4. Discussion

Previous electron microscopic analyses have shown that collagen fibrils in patients with mcEDS-*CHST14* are dispersed in the reticular dermis, in contrast to the regularly and tightly assembled fibrils observed in healthy individuals [7,16]. In addition, patients show small-sized collagen bundles comprising collagen fibrils with variable diameters separated by irregular interfibrillar spaces [8]. In the present study, we established in vitro models of fibroblast-mediated collagen network formation that recapacitate mcEDS-*CHST14* pathology. Electron microscopic analysis revealed impaired fibrillar organization in mcEDS-*CHST14*-mimicking gels, similar to the abnormal collagen network observed in patients with mcEDS-*CHST14*.

There are two forms of mcEDS resulting from loss-of-function mutations in *CHST14* or *DSE* [2,3,4]. Both forms of mcEDS shares the primary diagnostic criteria, including multiple congenital contractures, characteristic craniofacial features, and characteristic cutaneous features. However, core skin features and joint manifestations are significantly less common in patients with mcEDS-*DSE* than in patients with mcEDS-*CHST14* [10]. In mice, *Dse*-deficient fibrils have a larger diameter than wild-type fibrils, while *Chst14*-deficient fibrils do not show such an increase [26]. These studies suggest that CHST14 and DSE deficiency have different impacts on the pathogenesis of mcEDS. Our in vitro collagen network formation may be useful for analyzing not only mcEDS-*DSE* but also other EDS pathologies.

Using decorin isolated from cultured skin fibroblasts of patients and *Chst14*^−/−^ mice, we indicated that decorin is responsible for pathological collagen fibrinogenesis. The decorin preparation used in this study may contain proteoglycans other than decorin. However, since decorin is the major carrier protein of DS chains in the dermis, the impaired collagen fibrillar organization in the mcEDS-*CHST14*-mimicking gels may be due to the abnormal glycosylation of decorin. Further studies using recombinantly expressed decorin may clarify the exact contribution of decorin in mcEDS.

Decorin, a member of small leucine-rich proteoglycans, localizes to the surfaces of collagen fibrils and plays pivotal functions in fibrillogenesis [19,27]. A central domain of decorin is composed of ten leucine-rich repeats flanked by two cysteine-rich regions at the amino- and carboxyl-terminus. The arch-shaped structure of decorin allows interaction with the triple helix of type I collagen molecule [27]. Amino acid sequences located between the fourth and sixth leucine-rich repeats mediate the interaction with collagen [28,29]. Decorin-deficient mice display aberrant organization of collagen fibrils and fragile skin with decreased strength and stiffness [30,31]. The fibril diameter in decorin-deficient mice is more heterogeneous than that in controls, similarly to what was observed in our mcEDS-*CHST14* models. Furthermore, an in vitro study also shows that the inclusion of decorin during collagen fibrillogenesis increases the elasticity and tensile strength of resulting collagen gels [32]. These reports are consistent with our conclusion that the abnormal collagen network in the mcEDS-*CHST14* models is due to a functional defect of decorin.

The amino-terminal domain of decorin contains a single DS chain. Since the protein core of decorin mainly mediates the interaction with collagen, it is not fully understood how the loss of DS chain affects collagen fibrinogenesis. However, emerging evidence has indicated a link between DS chains on decorin and the collagen network. In the shape module model, the protein core of decorin is associated with collagen fibrils, while DS chains interact with each other and form antiparallel duplexes that bridge adjacent collagen fibrils [33,34,35]. On the other hand, several lines of study have indicated that DS chains associate with collagen fibrils by electrostatic interaction but not from the antiparallel duplexes [36,37]. These two models are not mutually exclusive, and a recently proposed ring mesh model may explain the apparent discrepancy. In this model, multiple DS chains of decorin interact with each other around collagen fibrils and form a ring mesh-like structure [16,17,38]. Each ring surrounds a collagen fibril at its D-band and fuses with adjacent rings to create a planar network. In any case, DS chains on decorin affect the interfibrillar distance between the collagen fibrils and the overall structure of the collagen network.

The DS chains on decorin are replaced with a CS chain in the mcEDS-*CHST14* patient [7,16]. We also confirmed the replacement of GAG chains on decorin in *Chst14*-deficient mice. DS chains show unique conformational flexibility because IdoA residues are in an equilibrium of 1C4, 2S0, and 4C1 conformations. In contrast, the structure of CS chains is rigid due to the fixed 4C1 conformation of GlcA in CS chains [39]. Therefore, the replacement from DS to CS chains may reduce the flexibility of the GAG chains on decorin and destabilize the GAG-antiparallel duplex. Recent studies have revealed that DS chains on decorin display a curved shape and tightly adhere to the outer surface of collagen fibrils [16,17]. In *CHST14*-deficient humans and mice, CS chains on decorin linearly extend from collagen fibrils resulting in a disrupted ring mesh-like surrounding collagen fibrils and spatial disorganization of collagen networks. The present study also demonstrates that the alternation of GAG chains on decorin from DS to CS chains causes impaired collagen fibrinogenesis using in vitro reconstitution models.

It has been reported that the initial interaction between fibroblasts and collagen leads to rearrangement and increased density of the collagen fibrils [20,40]. Collagen gel contraction provides an in vitro model for wound healing, fibrosis, scar contraction, and connective tissue morphogenesis. We found a delayed gel contraction in mcEDS-*CHST14* models suggesting that DS supports the initial phase of collagen gel contraction. DS impacts cell proliferation, migration, and tissue morphogenesis through interactions with secreted molecules such as fibroblast growth factors and hepatocyte growth factors [13,41,42,43,44]. A disturbed pleiotropic function of DS in mcEDS-*CHST14* models may result in delayed gel contraction during the early phase of collagen network remodeling. 

In conclusion, this study established in vitro collagen network formation models of mcEDS-*CHST14* and revealed that the alternation of GAG chains on decorin from DS to CS chains leads to abnormal collagen assembly. This study may provide useful in vitro models of mcEDS-*CHST14* to elucidate the pathomechanism of this disease.

## Figures and Tables

**Figure 1 genes-14-00308-f001:**
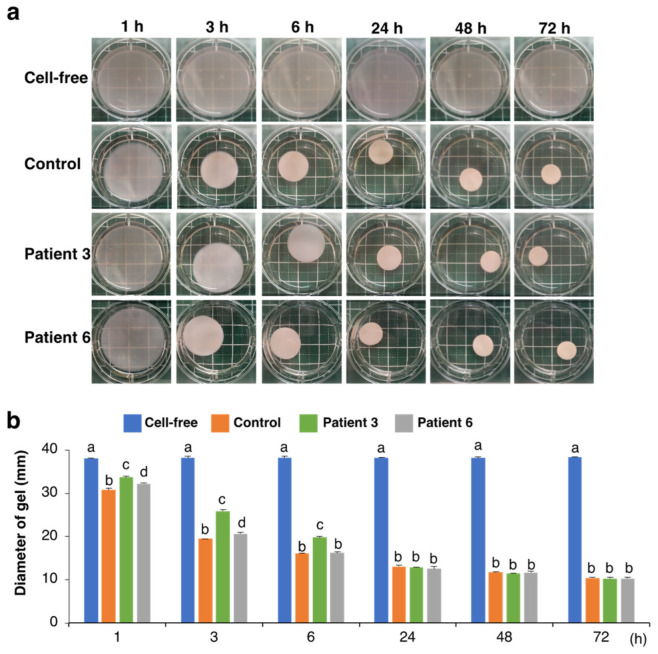
Effect of fibroblasts from mcEDS-*CHST14* patients on the contraction of type I collagen gels in vitro. (**a**) Fibroblasts derived from control and two patients were embedded in collagen gels and imaged at indicated time points. No contraction was observed in the absence of cells (Cell-free). (**b**) Diameter of the collagen gels. Different letters on the bars indicate significant differences (*p* < 0.01) from each other in all combinations based on multiple comparisons (Tukey–Kramer’s test). n = 3 for each group. Mean ± SE.

**Figure 2 genes-14-00308-f002:**
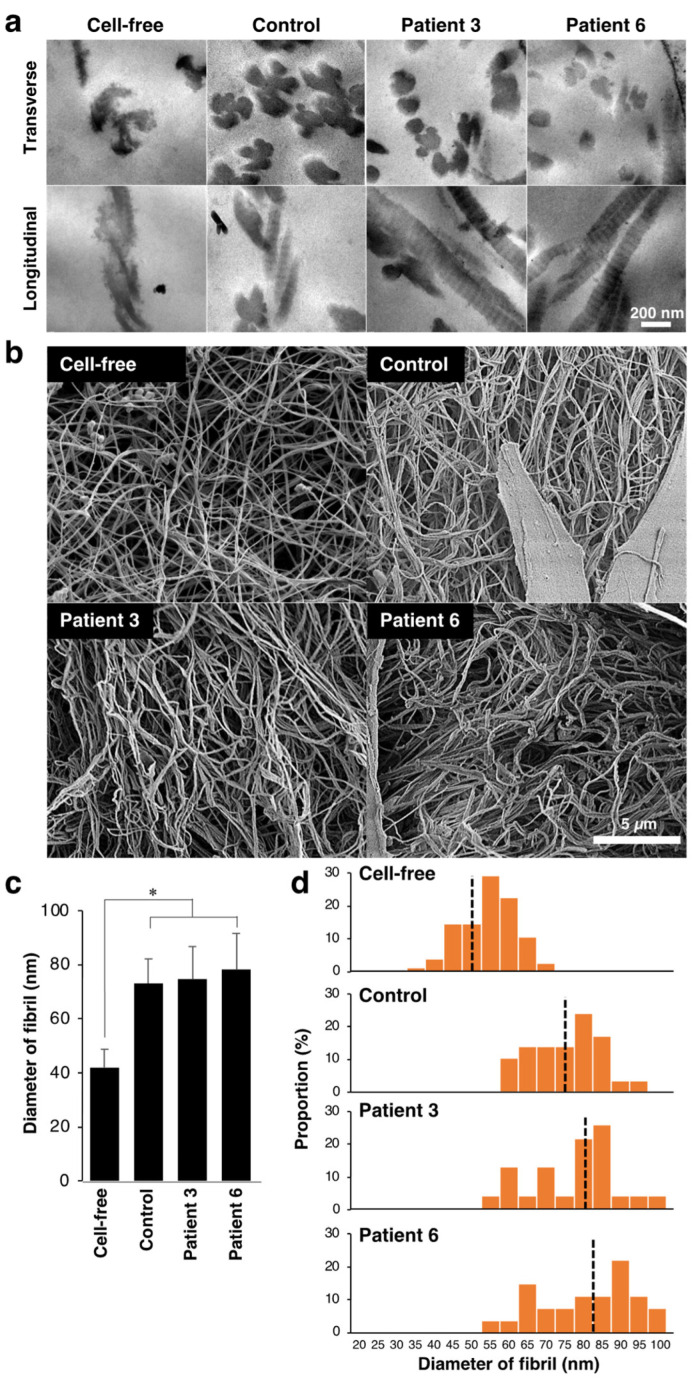
Effect of fibroblasts from mcEDS-*CHST14* patients on the fibrillar organization of type I collagen gels. (**a**) Transmission electron microscopy analysis of collagen fibrils in the cell-free, control, and patient-mimicking gels. Multiple fibrils are assembled to form collagen fibers in the control gel. (**b**) Scanning electron microscopy analysis of collagen fibrils. (**c**) The diameter of collagen fibrils. n = 75 fibrils for Cell-free, n = 29 for control, n = 23 for patient 3, n = 27 for patient 6. Mean ± SE; * *p* < 0.01; Tukey–Kramer’s test. (**d**) Histogram showing the distribution of the fibril diameter size. The Black dotted line is the median.

**Figure 3 genes-14-00308-f003:**
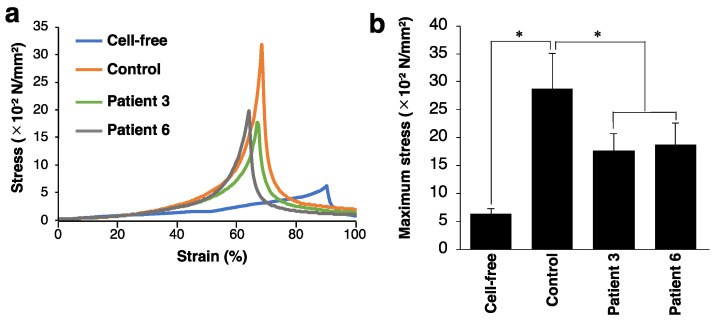
Effect of fibroblasts from mcEDS-*CHST14* patients on the mechanical strength of type I collagen gels. (**a**) Representative stress–strain curves of the cell-free, control, and patient-mimicking gels. (**b**) Bar graph showing the maximum stress of the gels. n = 8 gels for cell-free, n = 6 for control, n = 8 for patient 3, n = 8 for patient 6. Mean ± SE; * *p* < 0.01; Tukey–Kramer’s test.

**Figure 4 genes-14-00308-f004:**
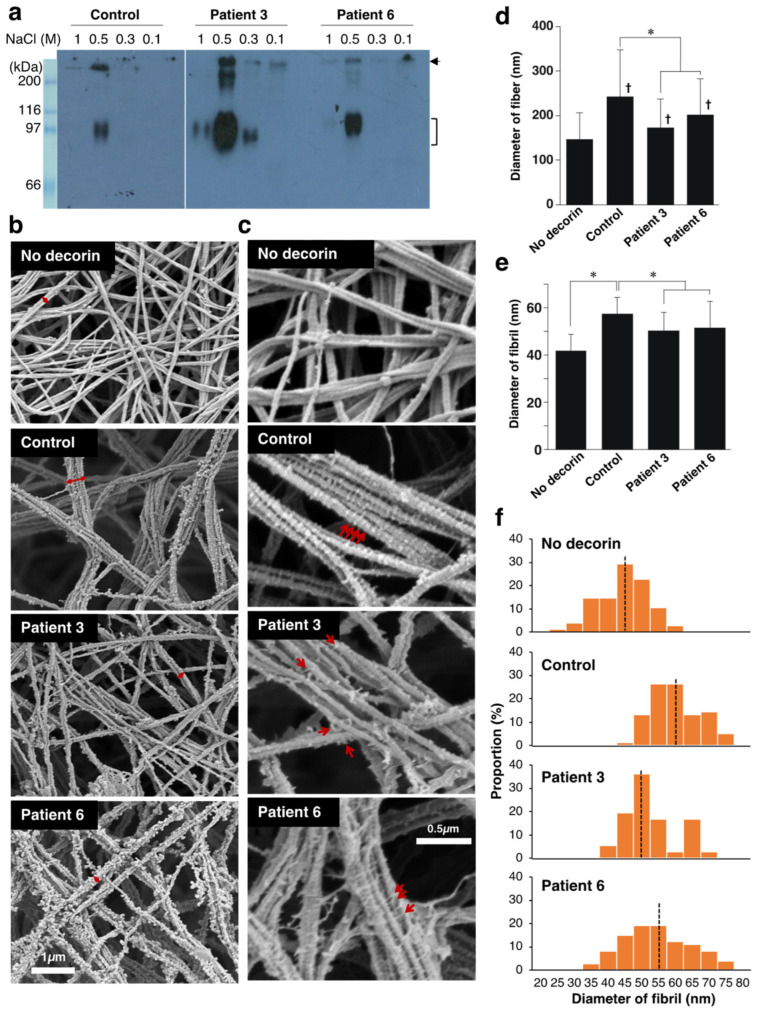
Effect of decorin isolated from normal or patient’s fibroblasts on the fibrillar organization of type I collagen in vitro. (**a**) Purification of decorin from cultured medium of control and patient’s fibroblasts. The 0.1, 0.3, 0.5, and 1.0 M NaCl fraction of anion exchange chromatography was subjected to immunoblotting to detect decorin. The bracket indicates the position of decorin. The arrow indicates unidentified components. (**b**) Scanning electron microscopy analysis of collagen fibers incubated with or without decorin. The red two-headed arrow shows collagen fibers composed of multiple fibrils. (**c**) Observation at higher magnification visualized GAG chains as indicated by the red arrows. (**d**,**e**) The diameter of collagen fibers. n = 211 for No-decorin, n = 124 for control, n = 121 for patient 3, n = 70 for patient 6. Mean ± SE; * *p* < 0.01; † *p* < 0.05 vs. No decorin; Tukey–Kramer’s test. (**e**) The diameter of collagen fibrils. n = 75 for No-decorin, n = 76 for control, n = 72 for patient 3, n = 74 for patient 6. Mean ± SE; * *p* < 0.01 vs. No decorin; Tukey–Kramer’s test. (**f**) Histogram showing the distribution of the fibril diameter size. The Black dotted line is the median.

**Figure 5 genes-14-00308-f005:**
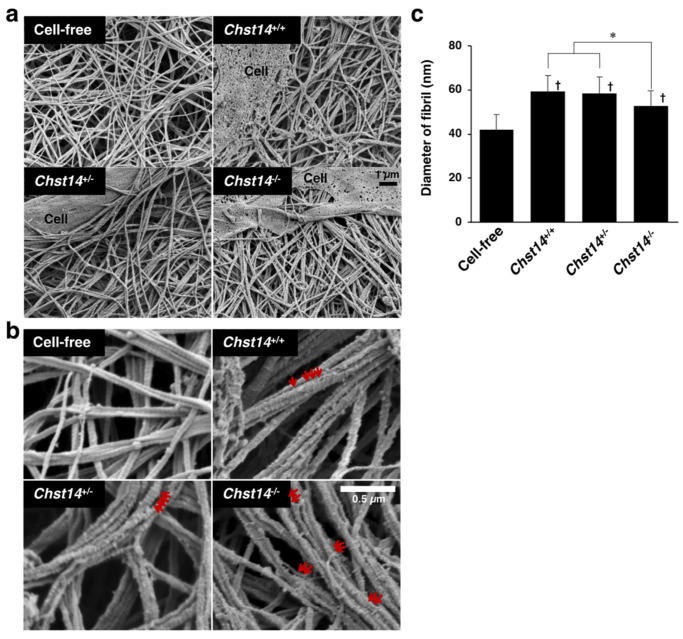
Effect of fibroblasts from *Chst14*-deficient mice on the fibrillar organization of type I collagen gels in vitro. (**a**) Scanning electron microscopy analysis of fibrils in the collagen gels co-cultured with or without fibroblasts from *Chst14*^+/+^, *Chst14*^+/−^, and *Chst14*^−/−^ mice. (**b**) Observation at higher magnification visualized GAG chains as indicated by the red arrows. (**c**) The diameter of collagen fibrils. n = 75 fibrils for Cell-free, n = 105 for *Chst14*^+/+^, n = 116 for *Chst14*^+/−^, n = 108 for *Chst14*^−/−^. Mean ± SE; * *p* < 0.01; † *p* < 0.05 vs. Cell-free; Tukey–Kramer’s test.

**Figure 6 genes-14-00308-f006:**
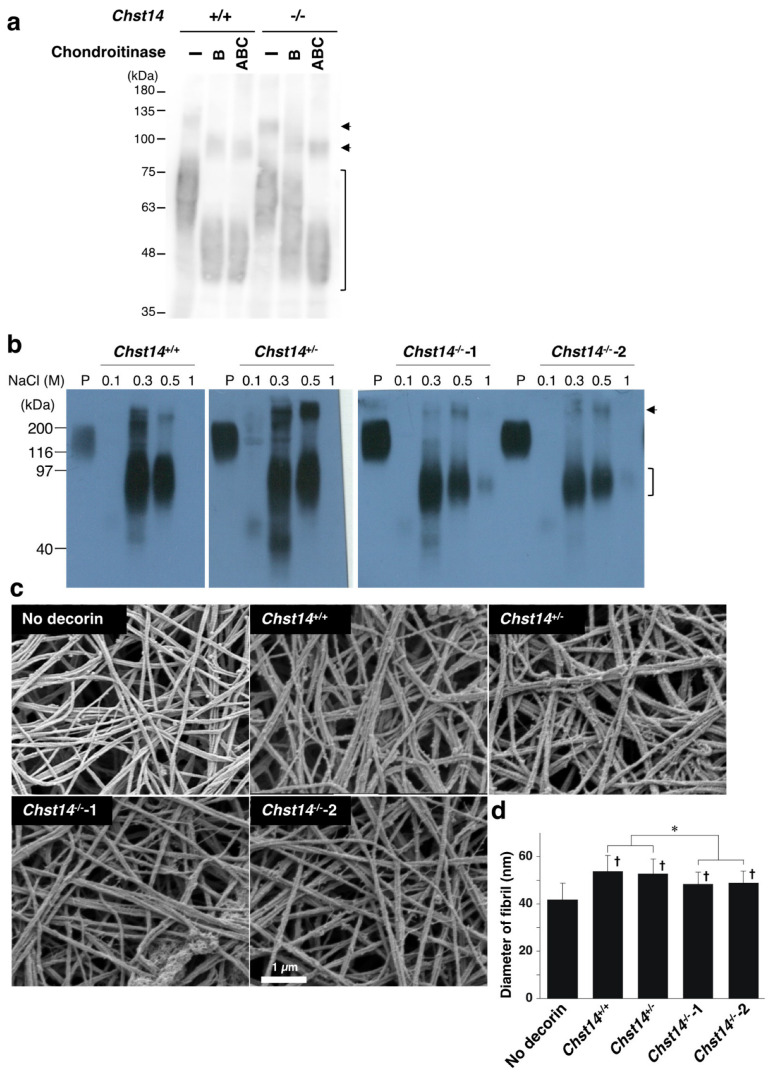
Effect of decorin isolated from *Chst14*^−/−^ mice on the fibrillar organization of type I collagen gels. (**a**) Immunoblot analysis of decorin in the skin lysate prepared from *Chst14*^+/+^ and *Chst14*^−/−^ mice after digestion with chondroitinase ABC or chondroitinase B. The bracket indicates the position of decorin. The arrows indicate unidentified components. (**b**) Purification of decorin from cultured medium of mouse fibroblasts with indicated genotype. The 0.1, 0.3, 0.5, and 1.0 M NaCl fraction of anion exchange chromatography was subjected to immunoblotting to detect decorin. Two different *Chst14*^−/−^ mice were used for purification of decorin. The bracket indicates the position of decorin. The arrow indicates unidentified components. P; pig placental decorin. (**c**) Scanning electron microscopy analysis of collagen fibers incubated with or without decorin isolated from mouse fibroblast with indicated genotype. (**d**) The diameter of collagen fibrils. n = 75 fibrils for No-decorin, n = 64 for *Chst14*^+/+^, n = 78 for *Chst14*^+/−^, n = 86 for *Chst14*^−/−^-1, n = 85 for *Chst14*^−/−^-2, Mean ± SE; * *p* < 0.01; † *p* < 0.05 vs. Cell-free; Tukey–Kramer’s test.

## Data Availability

Not applicable.

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
