# Peer review of "Collagen Network Formation in In Vitro Models of Musculocontractural Ehlers–Danlos Syndrome"

_genes, 2023, doi:10.3390/genes14020308_

Round 1

Reviewer 1 Report

The paper  Collagen network formation in in vitro models of musculocon-tractural Ehlers-Danlos syndrome genes 2098819 by Hashimoto at al is an interesting work where the effect of loss function of CHST14 on collagen fibrillation in collagen I  gels  is studied in in vitro models  using fibroblasts or decorin prepared from patients and targeted mice. The studies reveals impaired collagen fibrillation and weaker mechanical strength as earlier been published both in human and in mice with CHST14 deficiency.

Major comments

How are the dermal fibroblasts obtained.

It is unclear how pure the various decorin preparations are. In medium and in the skin there is in addition to decorin   versican, several heparan sulfate proteoglycans and hyaluronan.  In figure 3a there are components larger than decorin. This is also the case in 6a where there is a component after chondroitinase at 100 kDa. What is that? In fig 6b there is a clear indication of decorin at 80 kDa. What is that at 200 kDa? Further where is the arrow.

Further at the nice scanning EM what decorin is being used, 0.3 and 0.5 M eluate in figure 6.

Minor comments

In the introduction, it is claimed that the epimerases reaction of GlcUA til IdoA is reversible. This occurs only in in vitro. In vivo few of IdoA containing disaccharides are formed in the absence of CHST14. To form long stretches of IdoA contain disaccharides ChHST14 is needed to enhance IdoA formation on the enzyme level.

In methods chondroitinase AC is not mentioned but it is used in supplemental figure 1-

Fig 1b Significance –between what?

 In supplement data no IdoA-containing structure are noted. It could be pointed out that the IdoA is likely to occur in tetrasaccharide or longer formed after chondroitinase AC digestion. What is the nature of the unidentified peaks in supplement figure 1a.  The tetrasaccharide are not  attacked by chondroitinase B. What about the small peaks in position 3 in supplements figure 1b

 In the discussion section, the effect on collagen fibers in animals with targeted DSE could be mentioned. Possibly it could be discussed why DSE targeted animals have fibrils with larger diameter while those of CHST14 targeted animals does not show this increase.

Author Response

Thank you very much for the helpful comments on our manuscript (genes-2098819) entitled “Collagen network formation in in vitro models of musculo-contractural Ehlers-Danlos syndrome”. We revised the manuscript according to the Reviewers’ suggestions. Please find attached a word file of the point-to-point response to the reviewers' comments as well as the revised manuscript, in which all changes and corrections have been indicated in red.

Reviewer 2 Report

The manuscript by Hashimoto et al. described the effects of type I collagen fibril formation under the mutation or deficiency of CHST14 using in vitro systems. The authors performed experiments with human mcEDS-CHST14 patient fibroblasts, mouse Chst14 deficient fibroblasts, or purified decorin from these cells to investigate how commercially available bovine type I collagen forms fibrils in the presence of them and compared wild-type controls. The authors concluded that different GAG chains in decorin caused by mutations and deficiency of CHST14 impaired collagen fibril organization.

Major comment

The concept of the study is interesting, and experimental approaches are reasonable; however, the authors need to state their results and authors’ interpretations more clearly.

1) Overall, as experimental transparency, authors must add how many samples/molecules/measurements/experiments they used to generate figures. I could not find any information in this manuscript.

2) Figure 1b – what does a-d indicate?

3) Page 7, line 215: “the assembly of fibrils was less frequent” is unclear. How do you calculate percentages? In Figure 2a, patient samples have more fibrils in the transverse image.

4) Figure 4a – The decorin from patient 6 seems to migrate a little slower than others, while the authors stated indistinguishable. The samples with similar protein concentrations are loaded in the same gel if you want to compare the molecular weight, such as in Figure 6a.

5) The magnified image of red two-arrowheads and red arrow area is helpful for readers to see the difference in Figures 4b and c. The scale bars are hard to read in Figures 4b and c. Similar suggestions for Figures 5a (scale bar) and b.

6) Figure 6a – what is the upper band around 100 kDa in +/+ and -/- of ABC and B. Why don’t the authors mention the lower band as the same size of treated decorin exists in the -/- untreated sample?

7) Figure 6b - why does big placenta decorin migrate very slower than the purified decorin which author used for experiments? In Chst14+/+ and +/- samples, there is a band around 200 kDa instead of missing in -/-. The authors should explain it.

8) I could not find any sentences about Figures 6c and d.

9) Discussion is very poor. The authors did not discuss their findings, which should be revised.

Minor comment

1) In physiological conditions, the cells need ascorbic acid to produce robust collagen molecules. The authors did not add ascorbic acid during their cell culture. Does the production of SLRPs, including decorin, affect the absence of ascorbic acid?

2) Other SLRPs are also involved in collagen fibril formation, like decorin. Is there any possibility that the mutations or deficiency in CHST14 affect their GAG chain attachment?

3) Why don’t the authors test a similar experiment describing Figure 6a to purified decorin from patient fibroblasts which are used in Figure 4? Do they show a difference?

Author Response

(The authors gave the same response as above.)

Reviewer 3 Report

This is a study providing a novel in vitro analysis model for elucidating pathology of mcEDS-CHST14. mcEDS is a rare disease, and the establishment of a new model using patients derived cells would further understanding of the pathogenesis. The proposed in vitro collagen gel model seems useful, however comparison between current model vs previous in vivo or in vitro culture data are missing in results as well as in discussion. Therefore, the manuscript would be better served by clearly stating the usefulness of this model.

Compared to figure 2 vs figure 4-6, the collagen fibril diameter is larger with the patient’s cells in figure 2, but not in the others. The authors need provide some explanation.

Regarding figure 6, is there any evidence of DS to CS replacement in CHST14 revealed elsewhere? If this is the first data, the author should emphasize this with usefulness of this model. If not, it is a bit confusing, the precise explanation would be required.

As for discussion, since the pathogenesis of DS to CS replacement has been elucidated previously, it would be better to focus the discussion on establishing a novel in vitro analysis model. For example, comparing in vivo CHST14 skin TEM images vs current in vitro TEM images. It is also recommended to discuss the usefulness of this system for analyzing other EDS pathologies.

Regarding the discussion on gel contraction, it would be better to add more depth to the discussion.

Overall, the data is solid and new system seems beneficial, addressing the better interpretation and significance would make this manuscript better. 

Author Response

(The authors gave the same response as above.)

Round 2

Reviewer 1 Report

The paper is modified as suggested in my review

Reviewer 2 Report

The authors have satisfyingly addressed my questions.

Reviewer 3 Report

The authors have addressed the issues raised by the first revision. The revised manuscript has significant improvements as compared to the original version. I don’t have any other comments and concerns.